# Spatiotemporal Patterns of Light Pollution on the Tibetan Plateau over Three Decades at Multiple Scales: Implications for Conservation of Natural Habitats

Yihang Wang [1,2,3], Caifeng Lv [1,2], Xinhao Pan [1,2], Ziwen Liu [1,2], Pei Xia [4], Chunna Zhang [5,6,7] and Zhifeng Liu [1,2,5,*]

1   State Key Laboratory of Earth Surface Processes and Resource Ecology (ESPRE), Beijing Normal University, Beijing 100875, China
2   School of Natural Resources, Faculty of Geographical Sciences, Beijing Normal University, Beijing 100875, China
3   Tsinghua University High School-Wangjing, Beijing 100102, China
4   College of Urban and Environmental Sciences, Peking University, Beijing 100871, China
5   Academy of Plateau Science and Sustainability, People's Government of Qinghai Province, Beijing Normal University, Beijing 100875, China
6   Qinghai Province Key Laboratory of Physical Geography and Environment Process, College of Geographical Science, Qinghai Normal University, Xining 810008, China
7   Key Laboratory of Tibetan Plateau Land Surface Processes and Ecological Conservation (Ministry of Education), Qinghai Normal University, Xining 810008, China
*   Correspondence: zhifeng.liu@bnu.edu.cn

**Abstract:** Light pollution (LP), induced by human activities, has become a crucial threat to biodiversity on the Tibetan plateau (TP), but few studies have explored its coverage and dynamics. In this study, we intended to measure the spatiotemporal patterns of LP on the TP from 1992 to 2018. First, we extracted the annual extent of LP from time-series nighttime light data. After that, we analyzed its spatiotemporal patterns at multiple scales and identified the natural habitats and the species habitats affected by LP. Finally, we discussed the main influencing factors of LP expansion on the TP. We found that the LP area increased exponentially from 1.2 thousand km$^2$ to 82.8 thousand km$^2$, an increase of nearly 70 times. In 2018, LP accounted for 3.2% of the total area of the TP, mainly concentrated in the eastern and southern areas. Several national key ecological function zones (e.g., the Gannan Yellow river key water supply ecological function zone) and national nature reserves (e.g., the Lalu Wetland National Nature Reserve) had a large extent of LP. The proportion of LP area on natural habitats increased from 79.6% to 91.4%. The number of endangered species with habitats affected by LP increased from 89 to 228, and more than a quarter of the habitats of 18 endangered species were affected by LP. We also discovered that roadways as well as settlements in both urban and rural areas were the main sources of LP. Thus, to lessen LP's negative effects on biodiversity, effective measures should be taken during road construction and urbanization on the TP.

**Keywords:** Tibetan plateau; light pollution; urbanization; highway; railway; biodiversity; landscape sustainability

## 1. Introduction

LP refers to the phenomenon of diffuse light, reflected light and glare from modern urban buildings and night lighting that causes interference or negative effects on people, animals and plants [1,2]. An increasing number of studies show that LP can affect species feeding, sleep, migration, reproduction, navigation, communication, habitat selection and social interaction, resulting in a series of ecological and environmental issues, such as disturbance of circadian rhythm, limited survival and reproduction of species, and destruction of ecosystem structure [3–16]. At present, social and economic activities such

as global urbanization are accelerating the expansion of LP at an unprecedented speed and degree; LP is becoming a hot topic in the field of biodiversity conservation and sustainable development [17,18].

The Tibetan plateau (TP) is undergoing rapid urbanization and road construction, driven by the "Belt and Road" initiative, the "New Urbanization" plan and the "Western Development" strategy [19]. Just as the "Belt and Road" initiative drove the rapid development of Central Asia and other regions [20–23], the human footprint on the TP increased by 28.43% from 1990 to 2010, much higher than the world average (9%) [24]. Increased LP due to urbanization and road construction affects the quantity, quality and connectivity of natural habitats, thus posing a serious threat to biodiversity [25]. In addition, the ecosystems on the TP are fragile and have low resilience [26]. Once disturbed by human activities, they start to deteriorate rapidly [27]. To limit the influences of LP, it is essential to reveal the temporal and spatial patterns of LP and its impacts on ecological protection and sustainable development on the TP. However, there is still a lack of LP research focusing on the whole region of the TP.

Nighttime light (NTL) data provide an effective way to assess LP on the TP. There are two ways to measure LP using nighttime light data. One is to characterize LP directly from the spatial distribution of areas exposed to light at nighttime. Using this approach, Fan et al. [28] analyzed the impact of global LP on terrestrial nature reserves and wilderness areas based on Defense Meteorological Satellite Program-Operational Linescan System (DMSP-OLS) nighttime light data. Kumar et al. [29] analyzed the change in LP in India based on DMSP-OLS nighttime light data. Another is to build models to assess LP based on night light data and ecological indicators. For example, Koen et al. [18] analyzed the level of threat to biodiversity from LP during 1992–2012 based on DMSP-OLS nighttime light data, using the number of pixels and brightness value of night light in a 20 km × 20 km quadrate as an LP index. Cabrera-cruz et al. [30] used NPP-VIIRS nighttime light data to establish the relationship between nighttime light and bird migration routes and analyzed the impact of LP on migratory bird migration. The first method is simpler and more feasible and can fully reflect the characteristics of the spatial and temporal patterns of regional LP.

In existing studies, the two widely used sources of nighttime light data, i.e., DMSP-OLS and Suomi National Polar-Orbiting Partnership-Visible Infrared Imaging Radiometer Suite (NPP-VIIRS) have different archival times, which limits the time of most LP-related studies to a short period (before or after 2013), leading to a decrease in the spatiotemporal continuity of related studies [29,31]. To improve the temporal extent of nighttime light data, Li et al. [32] recently created a long time series nighttime light dataset GHNTL (Harmonization of DMSP and VIIRS nighttime light data from 1992–2018 at the global scale) by integrating the two types of nighttime light data, providing new data to support the study of the spatiotemporal pattern of LP and its impact on the TP in the last 30 years.

This study aimed to analyze the spatiotemporal patterns of LP and its impacts on the TP based on the GHNTL nighttime light data. First, we quantified the distribution and dynamics of LP on the TP during 1992–2018. We quantified LP at four scales: ecoregion, national key ecological function zone, national nature reserve and national park. Second, we analyzed the effects of LP on various natural habitats and habitats of different species on the TP. Then, we analyzed the reasons for LP and gave policy suggestions to deal with LP on the TP.

## 2. Study Area and Materials

### 2.1. Study Area

There are 27 ecoregions, 10 national key ecological function zones, 46 national nature reserves and 4 national parks on the TP (Figure 1). According to incomplete statistics by the International Union for the Conservation of Nature [33], there are at least 1714 species on the TP, including 388 species of mammals, 1050 species of birds, 117 species of reptiles and 159 species of amphibians.

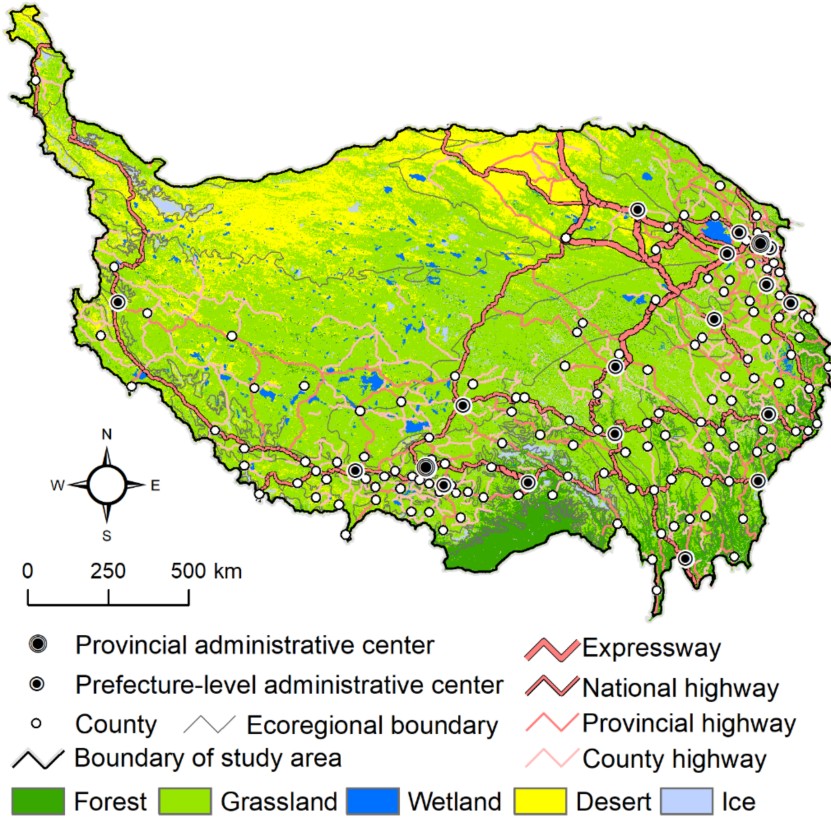

**Figure 1.** Study area. Note: the land cover information was obtained from 2018 version of ESACCI global land cover product.

The TP has undergone rapid social and economic development in recent years. The urbanization level of Qinghai province increased from 34.76% in 2000 to 55.52% in 2019, an increase of 21.8%. The urbanization level of the Tibet Autonomous Region increased from 18.93% in 2000 to 31.54% in 2019, an increase of 12.6% [34].

*2.2. Data*

NTL data obtained from the GHNTL [32] were used to analyze the LP over the TP. This dataset contains DMSP-OLS NTL time series data calibrated from 1992 to 2013 and DMSP-like NTL time series data transformed from NPP-VIIRS NTL from 2014 to 2018, with a spatial resolution of 1 km.

The land cover data from 1992 to 2018 used to assess the impact of LP on natural habitats were derived from the ESACCI global land cover product (http://maps.elie.ucl.ac.be/CCI/viewer/index.php) (accessed on 5 November 2022), with a spatial resolution of 300 m. The data include 22 land cover types. Referring to He et al. [35] and Zalles et al. [36], we adopted the habitat classification of the IUCN to extract natural habitats from ESACCI global land cover.

The species data used in this study were obtained from the IUCN Red List of Threatened Species (https://www.iucnredlist.org/resources/spatial-data-download) (accessed on 1 June 2021). Species data can be divided into four categories: mammals, birds, reptiles and amphibians. Each species was classified as Data Deficient (DD), Least Concern (LC), Near Threatened (NT), Vulnerable (VU), Endangered (EN), Critically Endangered (CR) and other categories according to the degree of endangered (Table 1). In addition, referring to Koen et al. [18], a separate division of nocturnal animals was summarized in this study.

**Table 1.** Number of species on the TP.

| Category (Abbreviation) | Mammals | Birds | Reptiles | Amphibians | Nocturnal Species | Total |
|---|---|---|---|---|---|---|
| Data Deficient (DD) | 24 | 2 | 8 | 26 | 7 | 60 |
| Least Concern (LC) | 285 | 931 | 100 | 91 | 103 | 1407 |
| Near Threatened (NT) | 29 | 52 | 1 | 16 | 6 | 98 |
| Vulnerable (VU) | 28 | 41 | 4 | 17 | 6 | 90 |
| Endangered (EN) | 20 | 15 | 3 | 7 | 0 | 45 |
| Critically Endangered (CR) | 2 | 8 | 1 | 2 | 0 | 13 |
| **Total** | **388** | **1050** | **117** | **159** | **122** | **1714** |

The ecoregion boundaries used in this study were derived from the Worldwide Fund for Nature (WWF) World Terrestrial ecoregion database [37] (http://worldwildlife.org/publications/terrestrial-ecoregions-of-the-world) (accessed on 1 June 2021). There are a total of 27 ecoregions on the TP. The boundary of national key ecological function zones was derived from the National and Regional Planning for Main Functional Zones [38]. There are 10 national key ecological function zones on the TP. The national nature reserve boundaries were derived from the Resources and Environment Science Data Center, and there are a total of 46 national nature reserves on the TP (http://www.resdc.cn/data.aspx?DATAID=272) (accessed on 1 June 2021). National park boundaries were derived from vectorization of national park planning maps [27]. There were four national parks on the TP: Giant Panda National Park, Sanjiangyuan National Park, Qilian Mountain National Park and Pudacuo National Park. The glacier dataset was provided by the National Cryosphere Desert Data Center [39] (http://www.ncdc.ac.cn) (accessed on 1 June 2021). The data of built-up areas used for LP attribution analysis came from the National Data Center for TP Science [40]. The basic geographic information came from the China National Basic Geographic Information Center, including 1:1,000,000 administrative boundaries, administrative centers, roads, settlements, rivers and lakes (http://ngcc.sbsm.gov.cn) (accessed on 1 June 2021).

## 3. Methods

### 3.1. Extraction of the LP Range

First, referring to Koen et al. [18], the night light data used in this study were comprehensively corrected to extract the LP range year by year from 1992 to 2018 (Figure 2). The value of night light from human activities is above 6.5 [32], and pixels with values lower than 6.5 are mostly abnormal values caused by the reflection of moonlight from glaciers or lakes. Therefore, the value of 6.5 was selected as the standard to binarize the original nighttime light data to remove abnormal values from the dataset.

Second, to further eliminate errors, the nighttime light range was corrected based on glacier and settlement data. As nighttime light patches should contain at least one residential area, nighttime light patches without residential areas were removed. In addition, patches of nighttime light containing glaciers were also removed.

Then, as the intensity of human activities on the TP is increasing year by year, we assumed that the range and intensity of nighttime light continuously increase. Referring to Liu et al. [41], based on this hypothesis, interannual correction was carried out from 1992 to 2018. After the above corrections, the areas exposed to nighttime light were extracted as the LP ranges [28,29].

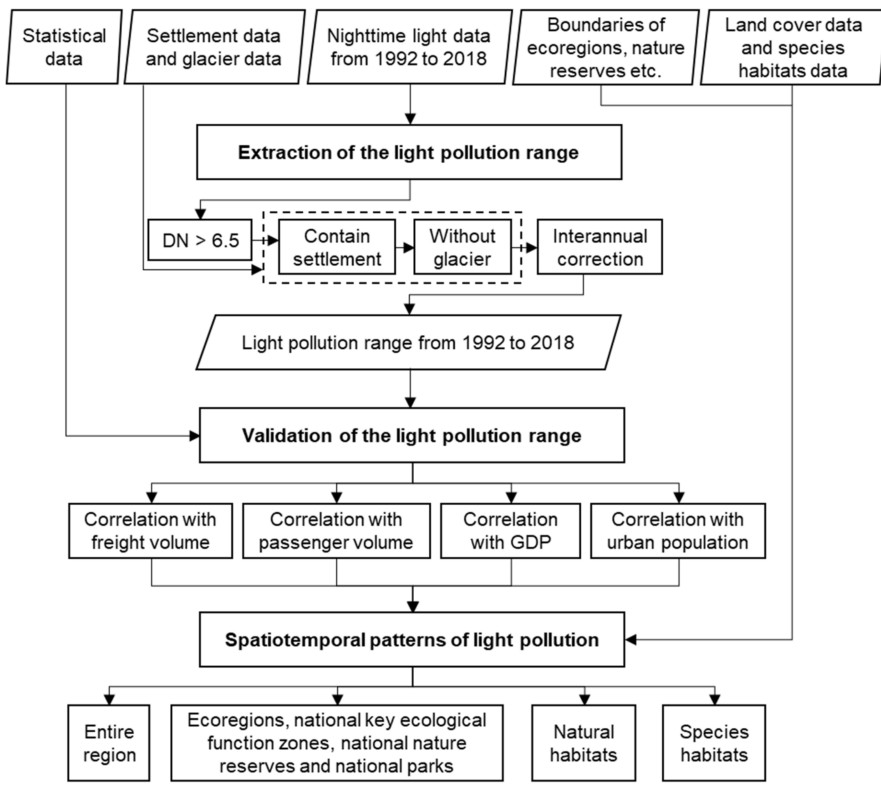

**Figure 2.** Flow chart.

### 3.2. Validation of the LP Range

Referring to Li et al. [42], provincial and prefecture-level socioeconomic indicators closely related to LP were selected. The accuracy of the annual LP ranges in Qinghai Province and the Tibet Autonomous Region during 1992–2018 were evaluated by the correlation analysis method. As the main sources of nighttime LP on the TP are roads and towns, four indices were selected: freight volume, passenger volume, GDP in secondary and tertiary industries and urban population. Freight volume and passenger volume represent the level of transportation development in this region, while GDP in secondary and tertiary industries and urban population represent the level of urbanization in this region.

### 3.3. Analysis of Spatiotemporal Patterns of LP

First, referring to Liu et al. [43], the landscape indices were used to quantify the spatial pattern of LP on the TP during 1992–2018. We selected total area (TA), patch density (PD), landscape shape index (LSI), aggregation index (AI) and largest patch index (LPI). PD refers to the number of patches per unit area, reflecting the degree of fragmentation or dispersion of light-polluted areas. The LSI is the normalized ratio of the patch perimeter to the patch area, which is an index used to measure the shape complexity of LP patches. The AI measures the concentration of LP patches. The LPI is the largest polluted patch on the entire TP. All indices were calculated by Fragstats software (V4.2) [44].

Second, referring to Fan et al. [28], Williams et al. [45] and Mu et al. [46], we analyzed the impact of LP expansion on the TP during 1992–2018 at the scale of ecoregions, national key ecological function zones, national nature reserves and national parks. We measured the degree of LP based on the proportion of LP area to the total area of each subregion. Since a threshold for defining the high-level LP at such large scales cannot be found, we used the approach adopted by Koen et al. [18] and He et al. [35], i.e., the regional average value, to extract the important areas facing relatively high-level LP.

Finally, referring to Koen et al. [18], He et al. [35] and McDonald et al. [47], we quantified the impacts of LP on biodiversity on the TP during 1992–2018 based on spatial overlay analysis and statistical analysis of the LP range with different types of natural

habitats and species habitats. To understand the effects of LP on natural habitats, we calculated the area of LP that overlapped with each type of natural habitats and calculated the proportion of this area to the total area of LP range year by year. The effects of LP on the habitats of different species were quantified from two aspects: one is the number of species with potential habitats affected by LP (i.e., species whose habitats overlap with LP areas), and the other is the proportion of light-polluted habitat area of the species. Additionally, the endangered species with high proportion of light-polluted habitat area were identified. To highlight key messages, we selected the LP range at seven milestones (1992, 1995, 2000, 2005, 2010, 2015, and 2018) from annual LP range between 1992 and 2018 for the above analyses.

## 4. Results

### 4.1. Validation of the LP Range

LP area is significantly correlated with various socioeconomic indicators (Figure 3). Correlation coefficients were all above 0.8 and passed the significance test of 0.001. The correlation coefficient between LP and GDP in secondary and tertiary industries is the highest (R = 0.89), followed by freight volume (R = 0.88), passenger volume (R = 0.84) and urban population (R = 0.82).

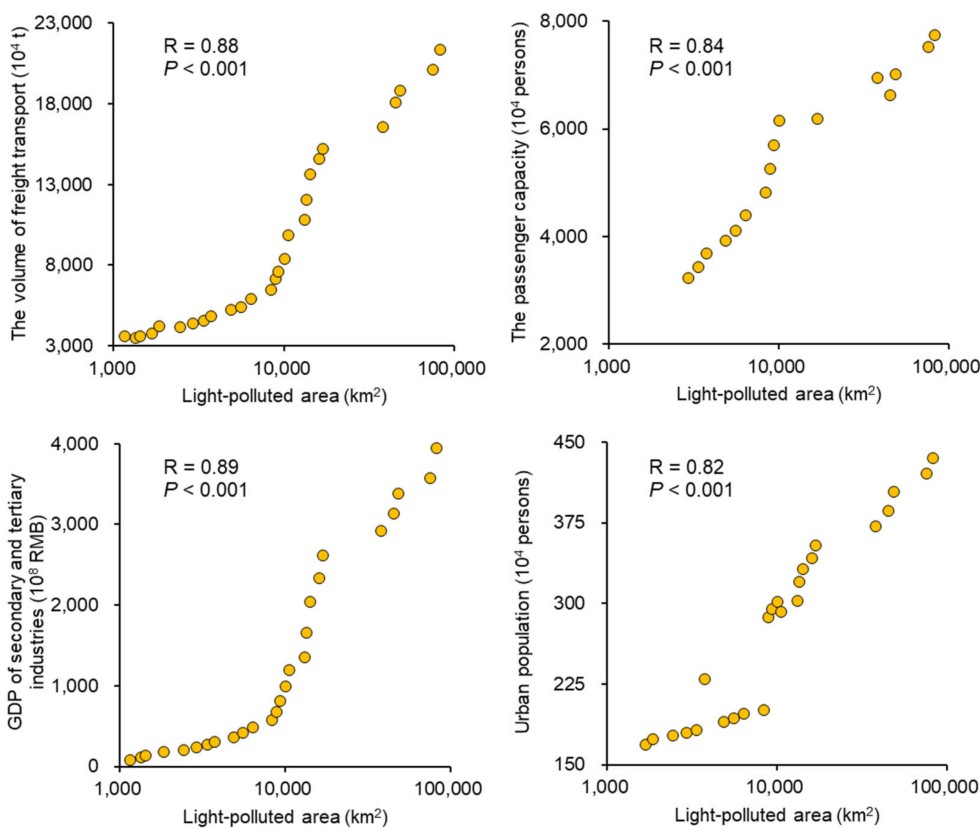

**Figure 3.** Accuracy evaluation of the light pollution range on the TP during 1992–2018.

### 4.2. Changes in LP on the TP from 1992 to 2018

From 1992–2018, the LP area on the TP increased exponentially. The LP area increased from 1.2 thousand $km^2$ to 82.8 thousand $km^2$, an increase of approximately 70 times, with an average annual growth rate of 30.6% (Figure 4a). In 2018, 3.2% of the total area of the TP was affected by LP. The LP in the eastern and southern TP was more serious and mainly occurred around urban and traffic routes. The LP in Xining and Lhasa was the most obvious (Figure 4). From 1992–2018, the degree of fragmentation in light-polluted areas on the TP gradually increased, and the PD increased from $1.6 \times 10^{-5}$ patches/$km^2$ in 1992 to

$3.8 \times 10^{-4}$ patches/km$^2$ in 2018, an increase of nearly 24 times (Figure 4b). The complexity of LP patches on the TP also gradually increased, and the LSI increased from 6.6 to 36.5 (Figure 4c). The concentration degree of LP showed an overall increasing trend, and the AI increased from 82.8 to 87.6 (Figure 4d). The LPI increased from $9.2 \times 10^{-5}$ to $4.1 \times 10^{-3}$, with an annual increase of 26.7% (Figure 4e).

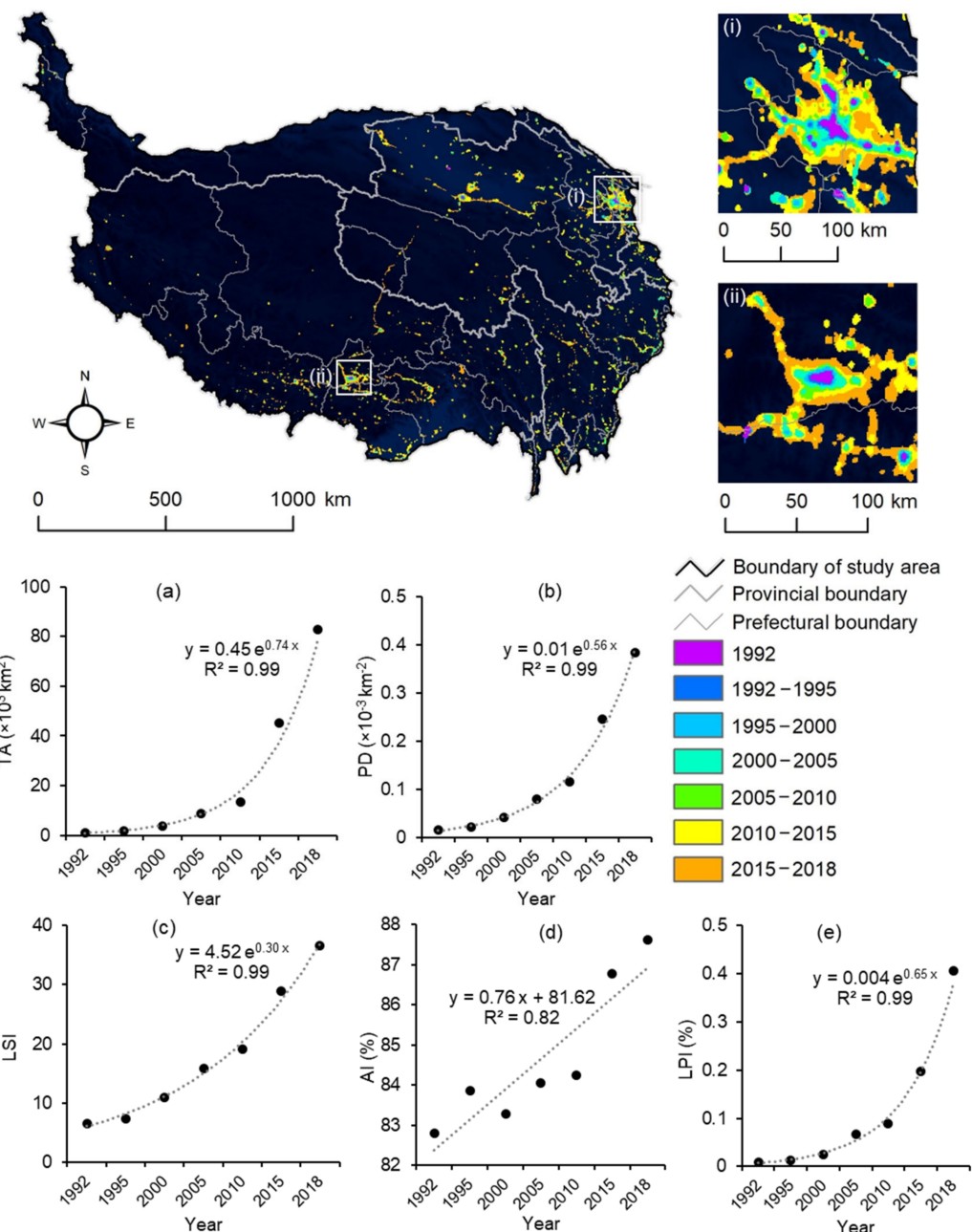

**Figure 4.** Changes in light pollution over the TP from 1992 to 2018: (**a**) total area (TA); (**b**) patch density (PD); (**c**) landscape shape index (LSI); (**d**) agglomeration index (AI); and (**e**) largest patch index (LPI) (Table S1).

### 4.3. Changes in LP at Different Scales

The results show that 26 of the 27 ecoregions on the TP were affected by LP, and 14 of them were affected by LP at a higher level than 3%, i.e., the average level of LP on the TP in 2018 (Figure 5a). The ecoregions in the southern TP were more affected, while the ecoregions in the northwest were less affected. In terms of time periods, the influence of LP



on the ecoregions of the TP has increased significantly since 2010 (Figure S1). Specifically, the proportion of LP area in five ecoregions, including Brahmaputra Valley semievergreen forests (No. 1 in Figure 5a), Yarlung Tsangpo arid steppe (No. 2) and Qionglai-Minshan conifer forests (No. 3), accounted for 12% or more. It should be noted that the most affected ecoregion was the Brahmaputra Valley semievergreen forest, located on the southern edge of the TP, where 32.90% of the entire region was affected by LP. The proportion of LP area in Qilian Mountain coniferous forests (No. 6), Qin Ling Mountain deciduous forests (No. 7) and Eastern Himalayan broadleaf forests (No. 8) accounted for 9—12%. The proportion of LP area in Hengduan Mountains subalpine coniferous forests (No. 9) and northeastern Himalayan subalpine coniferous forests (No. 10) was between 6% and 9%. The proportion of LP area in four ecoregions was between 3% and 6%: Eastern Himalayan subalpine conifer forests (No. 11), Southeast Tibet shrublands and meadows (No. 12), Nujiang Langcang Gorge alpine conifers and mixed forests (No. 13) and Qaidam Basin semi-deserts (No. 14).

All 10 national key ecological function zones on the TP were affected by LP, and four of them were affected by LP of more than the average level of LP on the TP in 2018 (Figures 5b and S2). These were the ecological function zone of the Gannan Yellow River water supply (No. 1 in Figure 5b), the forest and biodiversity in Sichuan and Yunnan (No. 2), the forest at the margin of the plateau in southeastern Tibet (No. 3), and the Zoige Grassland Wetland ecological function zone (No. 4). These national key ecological function zones were all located in the southeast of the TP. The national key ecological zones in the northwest of the TP were less affected by LP.

Thirty-four of the 46 national nature reserves on the TP were affected by LP, and 17 of them were affected by LP, accounting for more than the average level of LP on the TP in 2018 (Figures 5c and S3). Specifically, LP in the Lalu Wetland (No. 1 in Figure 5c), Lianhua Mountains (No. 2) and Xunhua Mengda Nature Reserve in Gansu (No. 3) accounted for 12% or more of the area. The area of LP in Wolong (No. 6) and Four Girls Mountain National Nature Reserve in Sichuan (No. 7) is between 9% and 12%. LP accounts for between 6% and 9% of the area of five national nature reserves, including the Haloxylon forest in Qinghai-Chaidam, Qinghai Province (No. 8), Baihe in Sichuan Province (No. 9) and Taizi Mountains in Gansu Province (No. 10). Five national nature reserves, including Sichuan Gexigou (No. 13), Gansu Duoer (No. 14) and Sichuan Gongga Mountain (No. 15), accounted for between 3% and 6% of the area of LP. It is worth noting that the Lalu Wetland National Nature Reserve, which was adjacent to Lhasa city and is mainly protected for its alpine wetland ecosystem, has been continuously affected by LP since 1992. By 2000, the range of LP affected the whole range of the reserve.

The impact of LP on the TP in 2018 was relatively small in national parks (Figure 5d). The area of LP in Giant Panda National Park (No. 1 in Figure 5d) accounts for 6.75% of its own area. and 2.22% of the source regions of the Yangtze River, Yellow River and Lancang River National Parks (No. 2) were affected by LP. Only 0.68% of the area of Qilian Mountain National Park (No. 3) was affected by LP, and Pudacuo National Park (No. 4) was not affected by LP. In terms of the time period, the impact of LP in Giant Panda National Park from 2000 to 2005 was greater, and, after 2010, the impact of LP in other national parks except Pudacuo Park increased significantly (Figure S4).

### 4.4. LP for Different Habitat Typess

The area of LP on natural habitats increased from 938 km$^2$ (79.6% of total LP area) in 1992 to 74.2 thousand km$^2$ (91.4% of total LP area) in 2018 (Table 2). The largest area of LP was found on grassland, accounting for 46.9% of LP area in 1992 and 54.7% of LP area in 2018 (Figure 6). The area of LP on forest is growing the fastest, and increased from 7.6% of LP area in 1992 to 26.0% of LP area in 2018 (Figure 6). Desert, wetland and ice accounted for 9.1%, 1.2% and 0.3% of the total LP area in 2018, respectively.

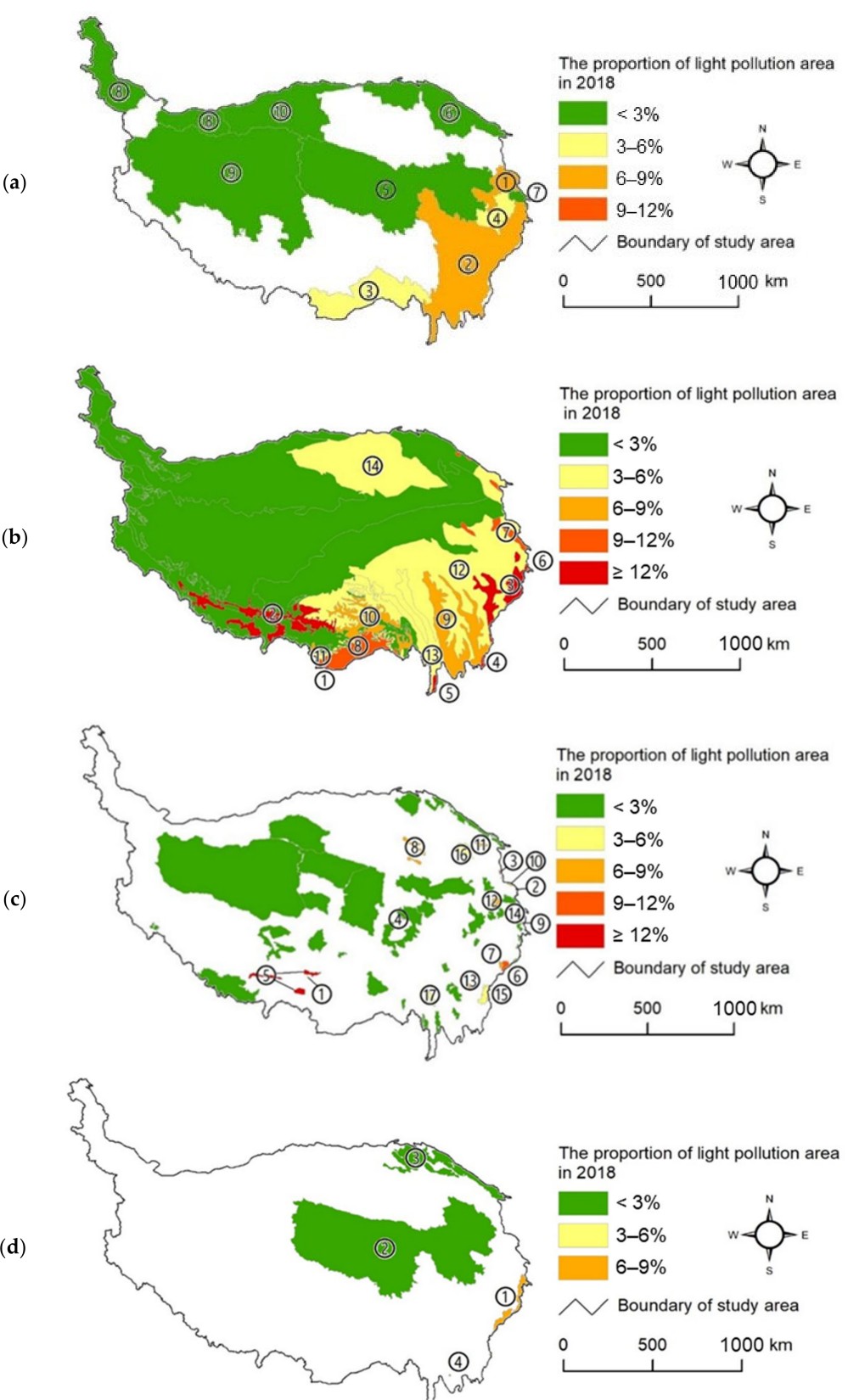

**Figure 5.** Light pollution in 2018 at the different scales. The names of labelled regions are shown in Figures S1–S4. (**a**) National key ecological function zone, (**b**) ecoregion, (**c**) national nature reserve, and (**d**) national park.

**Table 2.** Natural habitats affected by light pollution *.

| Natural Habitat | Year | | | | | | |
|---|---|---|---|---|---|---|---|
| | **1992** | **1995** | **2000** | **2005** | **2010** | **2015** | **2018** |
| Grassland | 552 (46.9%) | 855 (49.9%) | 1903 (50.5%) | 4583 (51.7%) | 6860 (51.1%) | 22,144 (49.9%) | 44,424 (54.7%) |
| Forest | 90 (7.6%) | 152 (8.9%) | 695 (18.4%) | 1928 (21.8%) | 3207 (23.9%) | 13,574 (30.6%) | 21,096 (26.0%) |
| Desert | 265 (22.5%) | 352 (20.6%) | 488 (13.0%) | 785 (8.9%) | 1096 (8.2%) | 3558 (8.0%) | 7395 (9.1%) |
| Wetland | 31 (2.6%) | 35 (2.0%) | 88 (2.3%) | 163 (1.8%) | 230 (1.7%) | 611 (1.4%) | 999 (1.2%) |
| Ice | 0 (0.0%) | 0 (0.0%) | 0 (0.0%) | 0 (0.0%) | 0 (0.0%) | 0 (0.0%) | 254 (0.3%) |
| **Sum** | **938 (79.6%)** | **1394 (81.4%)** | **3174 (84.3%)** | **7459 (84.2%)** | **11,393 (84.9%)** | **39,887 (89.8%)** | **74,168 (91.4%)** |

\* The number denotes the area of natural habitat affected by light pollution, while the proportion in parentheses denotes the percentage of natural habitat affected by light pollution to total light pollution area.

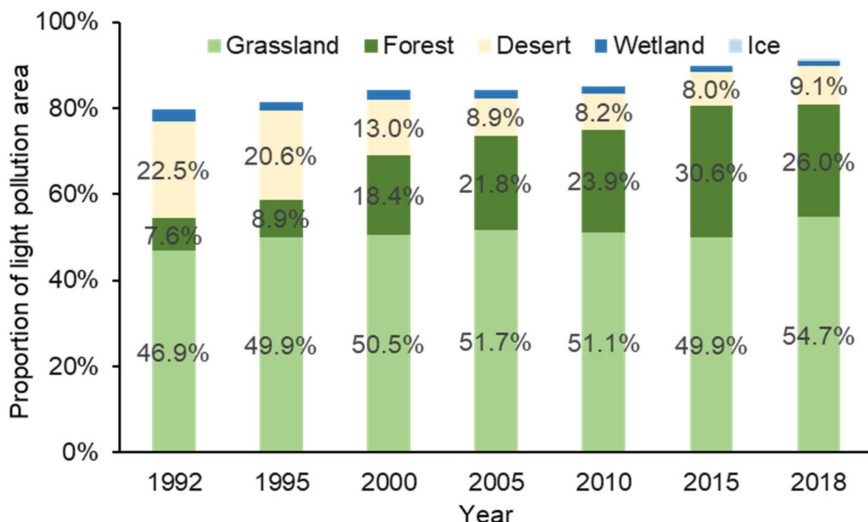

**Figure 6.** Natural habitats affected by light pollution.

*4.5. LP for Different Species Habitats*

The number of species with habitats affected by LP gradually increased on the TP during 1992–2018 (Figure 7a). Overall, the number of species with habitats affected by LP nearly doubled from 835 to 1619. The number of mammals with habitats affected by LP increased from 214 to 359, an increase of 0.7 times. The number of birds with habitats affected by LP increased from 532 to 1040, an increase of 0.9 times. The number of reptiles with habitats affected by LP increased from 52 to 110, more than doubling. The number of amphibians with habitats affected by LP increased by 2.7 times from 37 to 136. Among these, the number of nocturnal species with habitats affected by LP increased from 43 to 104, an increase of 1.4 times.

In addition, the number of endangered species with habitats affected by LP also increased year by year (Figure 7a). In total, the number of endangered species with habitats affected by LP increased from 89 to 228, an increase of approximately 1.5 times. The number of mammals affected increased 1.1 times, birds 1.6 times, reptiles 3.5 times, amphibians 2.8 times, and nocturnal animals approximately 2.7 times. Birds had the largest number of endangered species with habitats affected by LP (Figure 7a), while amphibians had the largest proportion of endangered species with habitats affected by LP (Figure 7b).

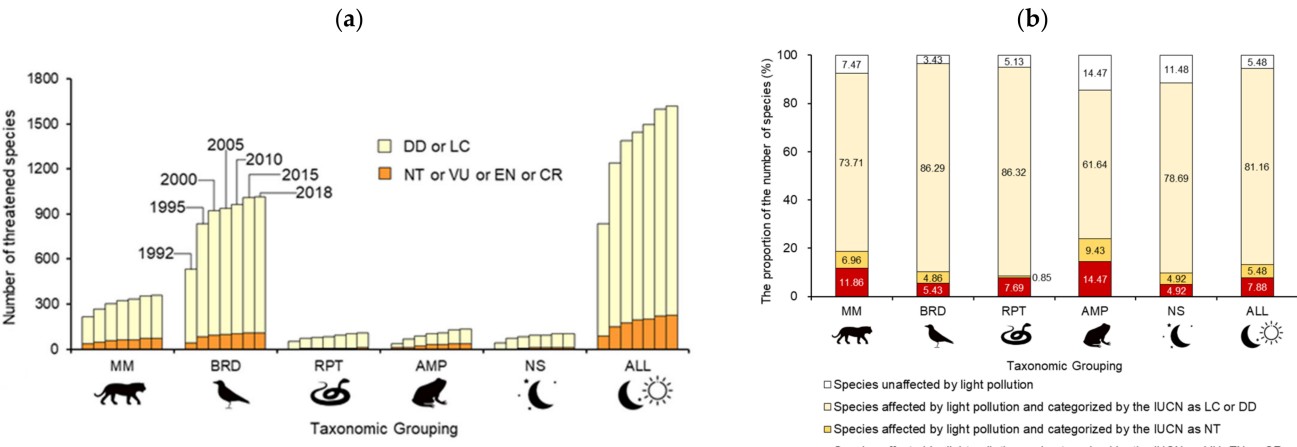

**Figure 7.** Different species habitats affected by light pollution on the TP. (**a**) Number of species with habitats affected by LP, 1992–2018 and (**b**) proportion of species with habitats affected by LP in 2018. Abbreviation: mammals (MM), birds (BRD), reptiles (RPT), amphibians (AMP), nocturnal species (NS).

The area of habitats affected by LP has gradually increased, especially since 2010 (Figure 8). The proportion of habitat area affected by LP for endangered species was generally larger than the average for all species. In 2018, a total of 18 endangered species, including 14 birds, 3 mammals and 1 reptile, were affected by LP, accounting for more than a quarter of their habitat area (Figure 8b; Tables S2 and S3).

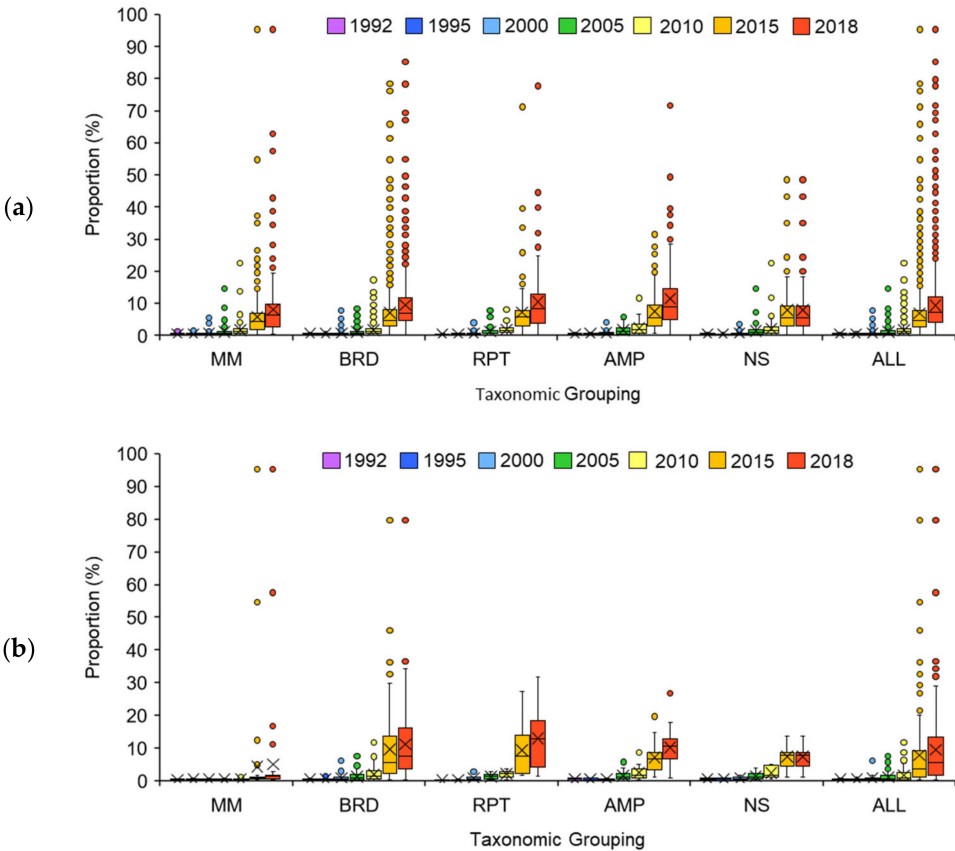

**Figure 8.** Proportion of habitat affected by light pollution on the TP. (**a**) Total species and (**b**) endangered species (i.e., NT, VU, EN and CR species).

Species richness increased from northwest to southeast, and endangered species also showed a similar pattern, which was consistent with the spatial distribution pattern of LP. According to the correlation analysis between the average level of species richness in different ecoregions and the percentage of increased area of LP from 1992 to 2018, the correlation coefficient between the percentage of increased area of LP and total species richness was 0.90 ($p < 0.001$) and that between the percentage of increased area of LP and endangered species richness was 0.99 ($p < 0.001$). The impact of LP on biodiversity, especially on endangered species, cannot be ignored.

## 5. Discussion

### 5.1. Main Reasons for the Increase in the LP Range

LP on the TP mainly occurs along traffic routes and around towns (Figures 1 and 4). We further conducted attribution analysis on the expanded range of LP on the TP from 1992 to 2018. Specifically, we analyzed the main sources of LP according to the distance of LP pixels from roads and settlements and LP pixels that had equal distances to roads and to settlements to the joint effects of both. The results show that the LP on the TP during 1992–2018 was mainly attributed to the influence of roads, accounting for 87.9% of the total area of new LP (Figure 9). In addition, 8.1% of the LP area was attributed to settlements. A total of 4.1% of the LP area was attributed to the combined influence of roads and settlements.

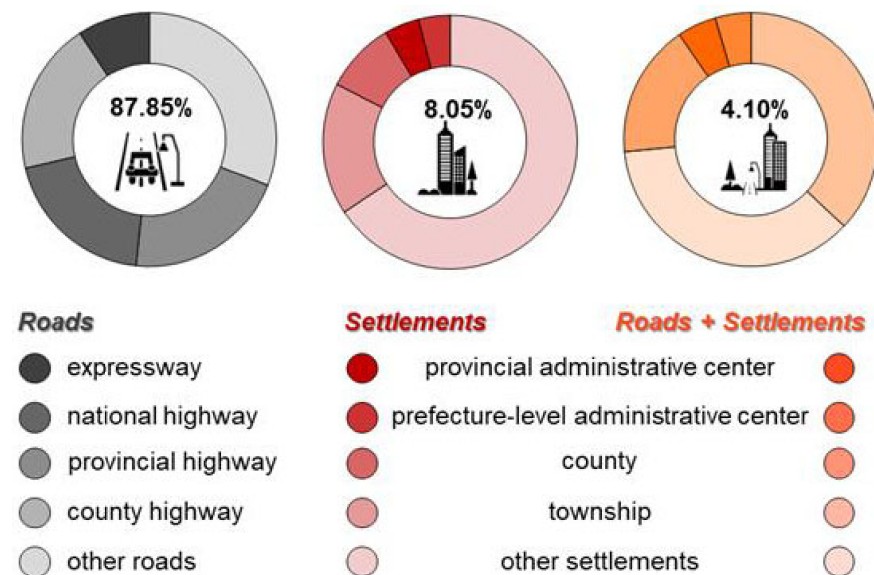

**Figure 9.** Attribution analysis of light pollution on the TP during 1992–2018.

### 5.2. Policy Implications

In recent years, the rapid increase in LP on the TP has posed a potential threat to nature reserves and endangered species. Therefore, it is suggested that LP should be reduced as much as possible in the future development process of the TP. Measures such as reducing lighting intensity, changing lighting spectral composition, and limiting lighting duration should be taken to effectively reduce LP [48]. For example, turning off an unnecessary artificial light source or adjusting the light source to the weakest brightness while still meeting the demand strictly limits the "blue" light that interferes with circadian rhythm and dark vision [49] and realizes adaptive lighting (intelligent switch) through technology. At the same time, the government should pay attention to the construction of "dark" ecological corridors, improve public awareness of LP, and strengthen relevant policy formulation and legislation (such as incorporating methods to avoid and mitigate LP in protected areas into management plans) [25,50,51]. Through the implementation of ecological compensation, the government can strengthen the protection and restoration of the ecological environment

within the scope of LP and take targeted development measures for different regions to achieve the coordinated development of regional ecological and environmental benefits, economic benefits and social benefits [52,53].

Considering that the impact of roads is the main factor for the increase in LP on the TP in recent years, it is particularly important to make overall planning in the future construction and development of roads. On the one hand, the impact of LP on the environment can be controlled by optimizing the planning of traffic networks. On the other hand, road light sources should be rationally arranged. The intensity of light sources should be strictly limited to reduce LP caused by roads, avoid damage to ecological corridors and curb adverse effects on biodiversity [54,55]. Future urban construction should also focus on the conservation of endangered species as much as possible (Tables S2 and S3), strictly restricting human activity within the territory of these species.

To promote effective measures for mitigating effects of LP on biodiversity conservation, we visually identified the major sources of LP for national nature reserves with relatively high-level LP based on the approach used in attribution analysis. In specific, both human settlements (including cities, towns, and villages) and roads were identified (Table 3). Although these areas caused a small proportion of the total LP, they had influences on the key natural habitats for biodiversity conservation. Therefore, measures should first be taken to control LP in these areas.

**Table 3.** Major source of light pollution for national nature reserves.

| ID in Figure 5c | Name | Abbreviation | Major Source of Light Pollution |
| --- | --- | --- | --- |
| 1 | Lalu Wetland | LW | Chengguan Distract in Lhasa city |
| 2 | Gansu Lianhua Mountain | GLM | Xiacheng town, Lianlu town |
| 3 | Xunhua Mengda | XM | Dahejia town |
| 4 | Longbao | Longbao | Longbao town |
| 5 | Black-necked crane of Brahmaputra River | BNCBR | Lhunzhub county, Maizhuokunggar county, Shigatse city, Road from Lhasa to Lhunzhub (G561), Road from Lhasa to Maizhuokunggar (G349) |
| 6 | Wolong | Wolong | Wolong town, Genda town |
| 7 | Four Girls Mountain | FGM | Siguniangshan town, Dawei town |
| 8 | Haloxylon forest in Qinghai-chaidam | HFIQC | Beijing-tibet highway (G6), Road from Beijing to Lhasa (G109) |
| 9 | Baihe | Baihe | Jiuhong road (S301) |
| 10 | Taizi mountains | TM | Songming town |
| 11 | The source region of rivers in the north of Datong | SRRND | Road from Xining to Zhangye (G227) |
| 12 | Gahai-zecha | GZ | Road from Gahai to Maqu (S204), Gaxiu village, Gahai village, Gongba village, Langmushi town |
| 13 | Gexigou | Gexigou | Yajing county, Xiangkezong |
| 14 | Duoer | Duoer | Road from Lianghekou to Maqu (S313) |
| 15 | Gongga Mountain | GM | Kangding city |
| 16 | Qinghai Lake | QL | Road from Beijing to Lhasa (G109) |
| 17 | Mangkang Yunnan snub-nosed monkey | MYSNM | Rumei town, Naxi nationality Town, Quzika Town |

Note: only the national nature reserves whose total light pollution range is above the average level of light pollution on the TP are listed.

*5.3. Future Perspectives*

In this study, the spatial-temporal pattern of LP on the TP during 1992–2018 was determined based on remotely sensed nighttime light data at multiple scales. The impacts of LP changes on natural habitat and species habitat were analyzed, therefore helping to fill the gap in the field of LP research on the TP. In the previous studies, Koen et al. [18] assessed the global LP from 1992 to 2012, while Li et al. [11] evaluated the LP for natural reserves in China in the same period. They both found that TP had relatively low LP between 1992 and 2012. However, the previous studies did not measure LP on the TP after 2012 in light of the inconsistence in NTL data. This study provided an assessment of LP over the past three decades based on the harmonized NTL data, and found that the LP had increased widely since 2012, posing threats to natural habitats and species habitats on the TP.

The shortcoming of this study is that we only considered the range of LP, and the intensity of LP was not fully considered. In fact, the farther away from the center of the light source, the weaker its ecological impacts. The degree of LP influence also depends on the duration of light and the weather conditions at that time. In addition, we only quantified the species with habitats covered by LP, and did not assess the effects of LP on species (e.g., reproduction and growth) on the TP. Therefore, we will further use nighttime light intensity information to assess the impact of LP on biodiversity on the TP in the future, and the field investigations into the effects of LP on species are also needed in order to reveal thresholds of coverage and intensity for LP that affect habitats.

## 6. Conclusions

LP has been increasing rapidly in recent years, which has caused a serious impact on the natural and species habitats in the region. From 1992–2018, the LP area of the TP increased from 1.2 thousand km$^2$ to 82.8 thousand km$^2$, an increase of approximately 70 times, which was mainly distributed in the east and south of the TP. Since 2010, the speed of LP coverage has significantly increased, and by 2018, the coverage of LP on the TP accounted for 3.22% of the total area. LP has seriously affected some ecoregions, national key ecological function zones, national nature reserves and national parks. The area of LP on natural habitats increased from 79.6% of total LP area to 91.4% of total LP area. The number of endangered species with habitats affected by LP rose from 89 to 228. Roads were the main source of LP, followed by urban and rural settlements. In the process of urbanization and road construction in the future, effective measures should be taken to control the scope and intensity of LP, focusing on endangered species affected by LP to promote biodiversity conservation.

**Supplementary Materials:** The following are available online at https://www.mdpi.com/article/10.3390/rs14225755/s1. Figure S1: Light pollution at the ecoregional scale; Figure S2: Light pollution at the scale of national key ecological function zone; Figure S3: Light pollution at the scale of national nature reserve; Figure S4: Light pollution at the national park scale; Table S1: Landscape index of light pollution area on the TP from 1992 to 2018; Table S2: Threatened species whose habitats are affected by light pollution by more than 25%; Table S3: Near threatened species whose habitats are affected by light pollution by more than 25%.

**Author Contributions:** Y.W. drafted the manuscript. Z.L. (Zhifeng Liu) conceived and guided this study. Y.W., C.L., X.P., Z.L. (Ziwen Liu), P.X. and C.Z. gave important advice on methodology and providing suggestions on the revision of the manuscript. All authors have read and agreed to the published version of the manuscript.

**Funding:** This work was supported by the Second Tibetan Plateau Scientific Expedition and Research Program (Grant No.2019QZKK0405) and the National Natural Science Foundation of China (Grant No. 41871185 & 41971271). It was also supported by the project from the State Key Laboratory of Earth Surface Processes and Resource Ecology, China.

**Institutional Review Board Statement:** Not applicable.

**Informed Consent Statement:** Not applicable.

**Data Availability Statement:** Not applicable.

**Conflicts of Interest:** The authors declare no conflict of interest.

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
