# Peer review of "Spatiotemporal Patterns of Light Pollution on the Tibetan Plateau over Three Decades at Multiple Scales: Implications for Conservation of Natural Habitats"

_remotesensing, doi:10.3390/rs14225755_

Round 1

Reviewer 1 Report

Review report

The manuscript entitled “Spatiotemporal patterns of light pollution on the Tibetan Plateau over three decades”, was submitted to Remote Sensing.

This manuscript attempts to assess spatiotemporal changes in light pollution (LP) in Tibetan Plateau over the past thirty years using a consistent dataset of nighttime light data. The key influences of LP on nature and the ecosystem were also included, which are useful for conservation strategies. Yet, the manuscript requires major revision and clarification before it could be published as a complete study.

The major problem of this manuscript is that LP was evaluated for a long-term period from 1992-2018. It should be noted that both LP and land cover are dynamic – they change year by year due to urban development (as authors stated in Section 4.1). Therefore, authors analyzed LP from 1992-2018 by overlaying with land cover in 2020 which is unreasonable. Land cover in 2018 versus 2020 is even different. For example, there were about 20% of forests affected by LP, however, due to deforestation for urban expansion, there were only 15% of forests remained affected by LP in 2018.

The detailed comments are given as follows:

1.     The title should be revised to emphasize the key contribution of this study such as ecosystem or species habitats.

2.     All numbers in the manuscript should be reconsidered and revised accordingly. For example, it should be 8.28 x 104 km2 instead of 8.28 x 104 km2 (Line 28). Besides, it is recommended that it uses the same format for easy comparison, e.g. 1.16 thousand km2 to 82.8 thousand km2.

3.     Introduction: “Belt and Road Initiative” (BRI) is an interesting point, which actually stimulates the corresponding regions somehow. Therefore, authors should give more examples of influences of BRI on urbanization and vegetation changes. For example, a few recent studies should be mentioned at least.

Jiang, Y., Lin, W., Wu, M., Liu, K., Yu, X., & Gao, J. (2022). Remote Sensing Monitoring of Ecological-Economic Impacts in the Belt and Road Initiatives Mining Project: A Case Study in Sino Iron and Taldybulak Levoberezhny. Remote Sensing, 14(14), 3308.

Nguyen, C. T., Chidthaisong, A., Limsakul, A., Varnakovida, P., Ekkawatpanit, C., Diem, P. K., & Diep, N. T. H. (2022). How do disparate urbanization and climate change imprint on urban thermal variations? A comparison between two dynamic cities in Southeast Asia. Sustainable Cities and Society, 82, 103882.

Song, Y., Aryal, J., Tan, L., Jin, L., Gao, Z., & Wang, Y. (2021). Comparison of changes in vegetation and land cover types between Shenzhen and Bangkok. Land Degradation & Development, 32(3), 1192-1204.

Fan, D., Ni, L., Jiang, X., Fang, S., Wu, H., & Zhang, X. (2020). Spatiotemporal analysis of vegetation changes along the belt and road initiative region from 1982 to 2015. IEEE Access, 8, 122579-122588.

4.     Methodology needs more clarification. Authors may add a general flowchart. Besides, the analysis of LP on habitat and species should be clarified. How did authors process these data?

5.     Authors should add trend lines and R-squared for each parameter.

6.     Line 199: If authors state that LP increased exponentially, they should show it in Figure 4, with an equation.

7.     Throughout section 4.3, authors mentioned different regions/zones by abbreviation on map and diagram. It should also be stated in the same form so that it is easy for the reader to follow. Also, the boundary of each region should be given in Supplementary.

Besides, a reference threshold of nighttime light, LP of higher than this value will affect the ecosystem, should be more suitable to compare rather than a value of areas higher (3%).

8.     Figure 11 should be revised. What is the meaning of different boxes?

9.     Figures and Tables: Authors are requested to check all the Figures and Tables.

Authors have to choose one of two, figure or table to illustrate their results. For example, in Figure 9, the pie chart and table are redundant. In addition, all figures need more explanation.

Please also check Table 1.

The data source of figure 1 should be stated.

Author Response

Point-by-Point Response

We would like to express our respect and gratitude to the editor and anonymous reviewers for their valuable comments on improving the quality of this manuscript.  We have carefully considered all the points raised by them. The following is our point-to-point responses in the order of their comments. Both a clear version and a “track change” version of the manuscript has been submitted. Our point-to-point responses are as follows.

Reviewer #1

Issue #1: The major problem of this manuscript is that LP was evaluated for a long-term period from 1992-2018. It should be noted that both LP and land cover are dynamic – they change year by year due to urban development (as authors stated in Section 4.1). Therefore, authors analyzed LP from 1992-2018 by overlaying with land cover in 2020 which is unreasonable. Land cover in 2018 versus 2020 is even different. For example, there were about 20% of forests affected by LP, however, due to deforestation for urban expansion, there were only 15% of forests remained affected by LP in 2018.

Response: Revised. We obtained the land cover data between 1992 and 2018 from ESACCI global land cover product (http://maps.elie.ucl.ac.be/CCI/viewer/index.php) and used these data to reanalyze the light pollution for natural habitats. Please refer to the second paragraph in Section 2.2 Data, Section 4.4, and Figure 6.

Issue #2: The title should be revised to emphasize the key contribution of this study such as ecosystem or species habitats.

Response: Revised. Please refer to the revised title: Spatiotemporal patterns of light pollution on the Tibetan Plateau over three decades at multiple scales: implications for conservation of natural habitats

Issue #3: All numbers in the manuscript should be reconsidered and revised accordingly. For example, it should be 8.28 x 104 km2 instead of 8.28 x 104 km2 (Line 28). Besides, it is recommended that it uses the same format for easy comparison, e.g. 1.16 thousand km2 to 82.8 thousand km2.

Response: Revised. Please refer to the revised manuscript.

Issue #4:  Introduction: “Belt and Road Initiative” (BRI) is an interesting point, which actually stimulates the corresponding regions somehow. Therefore, authors should give more examples of influences of BRI on urbanization and vegetation changes. For example, a few recent studies should be mentioned at least.

Jiang, Y., Lin, W., Wu, M., Liu, K., Yu, X., & Gao, J. (2022). Remote Sensing Monitoring of Ecological-Economic Impacts in the Belt and Road Initiatives Mining Project: A Case Study in Sino Iron and Taldybulak Levoberezhny. Remote Sensing, 14(14), 3308.

Nguyen, C. T., Chidthaisong, A., Limsakul, A., Varnakovida, P., Ekkawatpanit, C., Diem, P. K., & Diep, N. T. H. (2022). How do disparate urbanization and climate change imprint on urban thermal variations? A comparison between two dynamic cities in Southeast Asia. Sustainable Cities and Society, 82, 103882.

Song, Y., Aryal, J., Tan, L., Jin, L., Gao, Z., & Wang, Y. (2021). Comparison of changes in vegetation and land cover types between Shenzhen and Bangkok. Land Degradation & Development, 32(3), 1192-1204.

Fan, D., Ni, L., Jiang, X., Fang, S., Wu, H., & Zhang, X. (2020). Spatiotemporal analysis of vegetation changes along the belt and road initiative region from 1982 to 2015. IEEE Access, 8, 122579-122588.

Response: Revised. We added the references in the second paragraph in Introduction. Please refer to the revised manuscript.

Issue #5: Methodology needs more clarification. Authors may add a general flowchart. Besides, the analysis of LP on habitat and species should be clarified. How did authors process these data?

Response: Revised. The flowchart and the explanation were added. Please refer to Fig. 2 and the last paragraph in Section 3.3 in the revised manuscript.

Issue #6: Authors should add trend lines and R-squared for each parameter.

Response: Clarified and revised. In figure 3, we performed correlation analysis to validate the LP range, thus, only the R-coefficients were listed. In figure 4, we added trend lines and R-square to support our results.

Issue #7: Line 199: If authors state that LP increased exponentially, they should show it in Figure 4, with an equation.

Response: Revised. In figure 4, we added trend lines, R-square, and equations to support our results.

Issue #8:Throughout section 4.3, authors mentioned different regions/zones by abbreviation on map and diagram. It should also be stated in the same form so that it is easy for the reader to follow. Also, the boundary of each region should be given in Supplementary.

Besides, a reference threshold of nighttime light, LP of higher than this value will affect the ecosystem, should be more suitable to compare rather than a value of areas higher (3%).

Response: Clarified and revised. We modified the section 4.3 and figures related to this part. After revision, we used the serial numbers to connect main text and figures. In addition, the boundary of each region was uploaded in Supplementary.

We have double checked the relevant papers and cannot find a threshold for such large-scale analysis. Thus, we used the approach adopted by Koen et al. (2018) and He et al. (2014), i.e., the regional average value, to extract the important areas facing relatively high-level LP. We added this explanation in the second paragraph in Section 3.3 in the revised manuscript.

References:

  1. Koen, E.L.; Minnaar, C.; Roever, C.L.; Boyles, J.G. Emerging threat of the 21st century lightscape to global biodiversity. Global Change Biology 2018, 24 (6), 2315-2324.
  2. He, C.; Liu, Z.; Tian, J.; Ma, Q. Urban expansion dynamics and natural habitat loss in China: a multiscale landscape perspective. Global Change Biology 2014, 20 (9), 2886-2902.

Issue #9: Figure 11 should be revised. What is the meaning of different boxes?

Response: Revised. The different boxes denote the proportions of LP area for species habitats in different years. We added legends in this figure to explain it.

Issue #10: Figures and Tables: Authors are requested to check all the Figures and Tables.

Authors have to choose one of two, figure or table to illustrate their results. For example, in Figure 9, the pie chart and table are redundant. In addition, all figures need more explanation.

Please also check Table 1.

The data source of figure 1 should be stated.

Response: Revised. Please refer to the updated figures and tables in the revised manuscript.

Reviewer 2 Report

This manuscript “Spatiotemporal patterns of light pollution on the Tibetan Plateau over three decades” is investigating a very interesting issue regarding the spread of light pollution in Tibetan Plateau at different spatiial aggregation  from 1992-2018 using satellite data and its impact on species living in that region, however in my opinion, the study is not suitable for publication since it would need major revision.

Maybe I missed something or I did not understand properly and I apologize in this case, anyway, in my opinion, the study does not clearly explain the methodology that has been used to determine spatial and temporal LP and as a consequence spatila and temporal variation presented. In the manuscript it is not clear LP area (spatial temporal distribution is calculated. This is key information for this study, authors mentioned LP range but, honestly, I am not able to understand what they meant. In the study, even it is not clearly stated, it seems that LP is measured by DN. Did the authors use thresholds for defining LP?

Another key issue is the impact evaluation on species. First of all I think that  how the number of affected species is calculated should be more deeply explained. MY understanding is that the authors calculated the number of affected species just by counting the number of species present in the area without taking into consideration if they are concretely affected by light pollution and by intensity level of it. I think this is more and estimate of how many species are in lightnpollution conditions rather than affected by them, This takes more complex analysis in my opinion.

Other comments

Harmonize references through the paper (some are with names others with numbers)

Check the exponent format in the number like in lines 273-275 through the text

Introduction

In this section more references are needed “LP refers to the phenomenon of diffuse light, reflected light and glare from modern urban 44 buildings and night lighting that causes interference or negative effects on people, animals and Remote Sens. 2020, 12, x FOR PEER REVIEW 2 of 22 45 plants[1, 2]. An increasing number of studies show that LP can affect species feeding, sleep, 46 migration, reproduction, navigation, communication, habitat selection and social interaction, 47 resulting in a series of ecological and environmental issues, such as disturbance of circadian 48 rhythm, limited survival and reproduction of species, and destruction of ecosystem structure[3, 49 4].”

e.g.

Dimitriadis, C.; Fournari-Konstantinidou, I.; Sourbèsa, L.; Koutsoubas, D.; Mazaris, A.D. Reduction of sea turtle population recruitment caused by nightlight: Evidence from the Mediterranean region. Ocean. Coast Manag. 2018, 153, 108–115. [CrossRef]

Grubisic, M.; van Grunsven, R.H.A.; Kyba, C.C.M.; Manfrin, A.; Hölker, F. Insect declines and agroecosystems: Does light pollution matter?: Insect declines and agroecosystems. Ann. Appl. Biol. 2018, 173, 180–189. [CrossRef]

Dominoni, D.M.; Smit, J.A.H.; Visser, M.E.; Halfwerk, W. Multisensory pollution: Artificial light at night and anthropogenic noise have interactive effects on activity patterns of great tits (Parus major). Environ. Pollut. 2020, 256, 113314. [CrossRef] [PubMed]

Maggi, E.; Bongiorni, L.; Fontanini, D.; Capocchi, A.; Dal Bello, M.; Giacomelli, A.; Benedetti-Cecchi, L. Artificial light at night erases positive interactions across trophic levels. Funct. Ecol. 2020, 34, 694–706. [CrossRef]

Yang, Y.; Liu, Q.; Wang, T.; Pan, J. Light pollution disrupts molecular clock in avian species: A power-calibrated meta-analysis. Environ. Pollut. 2020, 265, 114206. [CrossRef] [PubMed]

 Ffrench-Constant, R.H.; Somers-Yeates, R.; Bennie, J.; Economou, T.; Hodgson, D.; Spalding, A.; McGregor, P.K. Light pollution is associated with earlier tree budburst across the United Kingdom. Proc. R. Soc. B 2016, 283, 20160813. [CrossRef]

 Škvareninová, J.; Tuhárska, M.; Škvarenina, J.; Babálová, D.; Slobodníková, L.; Slobodník, B.; Stˇredová, H.; Mind’aš, J.J. Effects of light pollution on tree phenology in the urban environment. Morav. Geogr. Rep. 2017, 25, 282–290. [CrossRef]

Bennie, J.; Davies, T.W.; Cruse, D.; Inger, R.; Gaston, K.J. Artificial light at night causes top-down and bottom-up trophic effects on invertebrate populations. J. Appl. Ecol. 2018, 55, 2698–2706. [CrossRef]

 Massetti, L. Assessing the impact of street lighting on Platanus x acerifolia phenology. Urban Urban Green 2018, 34, 71–77. [CrossRef]

Haim, A.; Abed, E.Z. Artificial light at night: Melatonin as a mediator between the environment and epigenome. Phil. Trans. R. Soc. B 2015, 370, 20140121. [CrossRef] [PubMed]

 Touitou, Y.; Reinberg, A.; Touitou, D. Association between light at night, melatonin secretion, sleep deprivation, and the internal clock: Health impacts and mechanisms of circadian disruption. Life Sci. 2017, 173, 94–106. [CrossRef] [PubMed]

Methods

Reading the paper I would be not able to find information how you classify the unit of area as LP or not LP to quantify spatial pattern. If this information is contained in some references, I would think that it could be relevant that is specified also in this manuscript since it is key information for interpreting the results.

Method lines 180-181:  I have a comment and a suggestion regarding “We measured the degree of LP based on the proportion of LP area to the total area of each subregion” I think this method does not exploit the full potential of detailed data on light pollution. Averaging over such large areas create large homogeneous areas with the same percentage of light pollution while probably is not the case thus not providing the best information for planners that need to know which are the sources and where to intervene (e.g. in area that has an average between 3 to 6%  there might be spots of higher light pollution and that are the places where to reduce it, and this is important information for managers that is available on the original dataset).

Another point, when you average regarding this is the light pollution has no boundaries so the light pollution in some pixels outside a region might affect the level of it.

Moreover you propose different maps of TP where an area have different level of light pollution according to the aggregation scale you use, and this might confuse the reader.

I would suggest to average LP in a different way rather than on the region, because it might drive to not effective actions.

Methods line 182-188 I believe that the description of this method should be improve and provide more details, rather than indicating references, because it regards a key objective of the study. How the impact is measure? Did you considered affected species just the number o species living in the area? How did you calculate it, did you use threshold of light pollution?

Results

Miss a map on the light pollution levels in 2018.

I would suggest the maps of percent of light pollution at different scales in one single figures (naming it a, b …) that makes easy to compare each other.

Fig 5, 6, 7 ,8 I would suggest to not use the same colors for the maps of spatial distribution in 2018  and the histograms of temporal increase (for each geographical analysis). Same colors with different meanings might confuse the reader  .

Fig. 5 ,6 ,7 ,8 and 10 have  several acronyms that are not defined in the captions, I understand that it makes the caption longer, but it is not easy to understand figures if acronyms definitions are in supplementary materials.

Discussion

Lines 336-347 this part of the discussion seems to me very general. I would suggest the authors to improve their discussion by focusing more on the results of this work.Maybe it could be better to compare this results with similar studies. Discuss more deeply the evaluation of the impact on species, since it is an investigation that requires more complex analysis and describe if there are any limitations in this sense for this study.

Author Response

Point-by-Point Response

We would like to express our respect and gratitude to the editor and anonymous reviewers for their valuable comments on improving the quality of this manuscript.  We have carefully considered all the points raised by them. The following is our point-to-point responses in the order of their comments. Both a clear version and a “track change” version of the manuscript has been submitted. Our point-to-point responses are as follows.

Reviewer #2

Issue #1: Maybe I missed something or I did not understand properly and I apologize in this case, anyway, in my opinion, the study does not clearly explain the methodology that has been used to determine spatial and temporal LP and as a consequence spatila and temporal variation presented. In the manuscript it is not clear LP area (spatial temporal distribution is calculated. This is key information for this study, authors mentioned LP range but, honestly, I am not able to understand what they meant. In the study, even it is not clearly stated, it seems that LP is measured by DN. Did the authors use thresholds for defining LP?

Response: Revised. The flowchart and following explanations were added to clarify our method for extracting the LP range. Please refer to Fig.2 and Section 3.1 in the revised manuscript.

Issue #2:  Another key issue is the impact evaluation on species. First of all I think that  how the number of affected species is calculated should be more deeply explained. MY understanding is that the authors calculated the number of affected species just by counting the number of species present in the area without taking into consideration if they are concretely affected by light pollution and by intensity level of it. I think this is more and estimate of how many species are in light pollution conditions rather than affected by them, This takes more complex analysis in my opinion.

Response: Revised. Yes, we just counted the number of species whose habitats overlap with LP areas. To avoid confusion, we revised Section 3.3 and Section 4.5 in the revised manuscript.

Issue #3:  Harmonize references through the paper (some are with names others with numbers)

Check the exponent format in the number like in lines 273-275 through the text

Response: Revised. Please refer to the revised manuscript.

Issue #4:  In this section more references are needed “LP refers to the phenomenon of diffuse light, reflected light and glare from modern urban 44 buildings and night lighting that causes interference or negative effects on people, animals and Remote Sens. 2020, 12, x FOR PEER REVIEW 2 of 22 45 plants[1, 2]. An increasing number of studies show that LP can affect species feeding, sleep, 46 migration, reproduction, navigation, communication, habitat selection and social interaction, 47 resulting in a series of ecological and environmental issues, such as disturbance of circadian 48 rhythm, limited survival and reproduction of species, and destruction of ecosystem structure[3, 49 4].”

e.g.

Dimitriadis, C.; Fournari-Konstantinidou, I.; Sourbèsa, L.; Koutsoubas, D.; Mazaris, A.D. Reduction of sea turtle population recruitment caused by nightlight: Evidence from the Mediterranean region. Ocean. Coast Manag. 2018, 153, 108–115. [CrossRef]

Grubisic, M.; van Grunsven, R.H.A.; Kyba, C.C.M.; Manfrin, A.; Hölker, F. Insect declines and agroecosystems: Does light pollution matter?: Insect declines and agroecosystems. Ann. Appl. Biol. 2018, 173, 180–189. [CrossRef]

Dominoni, D.M.; Smit, J.A.H.; Visser, M.E.; Halfwerk, W. Multisensory pollution: Artificial light at night and anthropogenic noise have interactive effects on activity patterns of great tits (Parus major). Environ. Pollut. 2020, 256, 113314. [CrossRef] [PubMed]

Maggi, E.; Bongiorni, L.; Fontanini, D.; Capocchi, A.; Dal Bello, M.; Giacomelli, A.; Benedetti-Cecchi, L. Artificial light at night erases positive interactions across trophic levels. Funct. Ecol. 2020, 34, 694–706. [CrossRef]

Yang, Y.; Liu, Q.; Wang, T.; Pan, J. Light pollution disrupts molecular clock in avian species: A power-calibrated meta-analysis. Environ. Pollut. 2020, 265, 114206. [CrossRef] [PubMed]

 Ffrench-Constant, R.H.; Somers-Yeates, R.; Bennie, J.; Economou, T.; Hodgson, D.; Spalding, A.; McGregor, P.K. Light pollution is associated with earlier tree budburst across the United Kingdom. Proc. R. Soc. B 2016, 283, 20160813. [CrossRef]

 Škvareninová, J.; Tuhárska, M.; Škvarenina, J.; Babálová, D.; Slobodníková, L.; Slobodník, B.; Stˇredová, H.; Mind’aš, J.J. Effects of light pollution on tree phenology in the urban environment. Morav. Geogr. Rep. 2017, 25, 282–290. [CrossRef]

Bennie, J.; Davies, T.W.; Cruse, D.; Inger, R.; Gaston, K.J. Artificial light at night causes top-down and bottom-up trophic effects on invertebrate populations. J. Appl. Ecol. 2018, 55, 2698–2706. [CrossRef]

 Massetti, L. Assessing the impact of street lighting on Platanus x acerifolia phenology. Urban Urban Green 2018, 34, 71–77. [CrossRef]

Haim, A.; Abed, E.Z. Artificial light at night: Melatonin as a mediator between the environment and epigenome. Phil. Trans. R. Soc. B 2015, 370, 20140121. [CrossRef] [PubMed]

 Touitou, Y.; Reinberg, A.; Touitou, D. Association between light at night, melatonin secretion, sleep deprivation, and the internal clock: Health impacts and mechanisms of circadian disruption. Life Sci. 2017, 173, 94–106. [CrossRef] [PubMed]

Response: Revised. We added these references. Please refer to the first paragraph in Section 1 in the revised manuscript.

Issue #5:  Reading the paper I would be not able to find information how you classify the unit of area as LP or not LP to quantify spatial pattern. If this information is contained in some references, I would think that it could be relevant that is specified also in this manuscript since it is key information for interpreting the results.

Response: Revised. We added Fig. 2 and revised Section 3.1 to highlight the threshold for extracting the LP range. In Koen et al. (2018), the threshold of 5.5 was used, while in the data document of GHNTL which was used in this study to extract the LP range, the threshold of 7 was recommended (Li et al., 2020). After manual verification and the validation based on statistical data, we finally used the threshold of 6.5.

References:

  1. Koen, E.L.; Minnaar, C.; Roever, C.L.; Boyles, J.G. Emerging threat of the 21st century lightscape to global biodiversity. Global Change Biology 2018, 24 (6), 2315-2324.
  2. Li, X.; Zhou, Y.; Zhao, M.; Zhao, X. A harmonized global nighttime light dataset 1992–2018. Scientific Data 2020, 7 (1).

Issue #6:  Method lines 180-181:  I have a comment and a suggestion regarding “We measured the degree of LP based on the proportion of LP area to the total area of each subregion” I think this method does not exploit the full potential of detailed data on light pollution. Averaging over such large areas create large homogeneous areas with the same percentage of light pollution while probably is not the case thus not providing the best information for planners that need to know which are the sources and where to intervene (e.g. in area that has an average between 3 to 6%  there might be spots of higher light pollution and that are the places where to reduce it, and this is important information for managers that is available on the original dataset).

Response: Revised. In this study, we followed Fan et al. (2019) and Kumar et al. (2019) and used the proportion of LP area to identify the relatively high-level light-polluted regions. As the reviewer states, this approach cannot find the LP hotspots and sources. To compensate for this shortcoming, we combined the key protected areas listed in results and the attribution analysis to visually interpret the major sources of LP. We only selected nature reserves because they are the most important areas for biodiversity conservation in China. Please refer to Table 2 and Section 5.2 in the revised manuscript.

References:

  1. Fan, L.; Zhao, J.; Wang, Y.; Ren, Z.; Zhang, H.; Guo, X. Assessment of Night-Time Lighting for Global Terrestrial Protected and Wilderness Areas. Remote Sensing 2019, 11 (22), 2699.
  2. Kumar, P.; Rehman, S.; Sajjad, H.; Tripathy, B.R.; Rani, M.; Singh, S. Analyzing trend in artificial light pollution pattern in India using NTL sensor's data. Urban Climate 2019, 27, 272-283.

Issue #7:  Another point, when you average regarding this is the light pollution has no boundaries so the light pollution in some pixels outside a region might affect the level of it.

Response: Revised. Similar to the previous question, when we identified the source of LP, we considered the effects of LP outside the protected area. Please refer to Table 2 and Section 5.2 in the revised manuscript.

Issue #8:  Moreover you propose different maps of TP where an area have different level of light pollution according to the aggregation scale you use, and this might confuse the reader.

I would suggest to average LP in a different way rather than on the region, because it might drive to not effective actions.

Response: Revised. Similar to Issue #6, to drive effective actions, we identified the major sources of LP in natural reserves. Controlling these sources of LP will effectively reduce the impacts of LP on habitats for biodiversity conservation. Please refer to Table 2 and Section 5.2 in the revised manuscript.

Issue #9:  Methods line 182-188 I believe that the description of this method should be improve and provide more details, rather than indicating references, because it regards a key objective of the study. How the impact is measure? Did you considered affected species just the number o species living in the area? How did you calculate it, did you use threshold of light pollution?

Response: Revised. The explanations were added to clarify our method. Please refer to Section 3.3 in the revised manuscript.

Issue #10:  Miss a map on the light pollution levels in 2018.

Response: Clarified. Since the light pollution range from 1992 to 2018 was interannual corrected, the extent with color in Fig. 4 can present the light pollution in 2018.

Issue #11:  I would suggest the maps of percent of light pollution at different scales in one single figures (naming it a, b …) that makes easy to compare each other.

Response: Revised. Please refer to the figure 5 in the revised manuscript.

Issue #12:  Fig 5, 6, 7 ,8 I would suggest to not use the same colors for the maps of spatial distribution in 2018  and the histograms of temporal increase (for each geographical analysis). Same colors with different meanings might confuse the reader  .

Response: Revised. We divided these figures to avoid confusion. Please refer to the figure 5 and figure s1-s4 in the revised manuscript.

Issue #13:  Fig. 5 ,6 ,7 ,8 and 10 have  several acronyms that are not defined in the captions, I understand that it makes the caption longer, but it is not easy to understand figures if acronyms definitions are in supplementary materials.

Response: Revised. The full names were added in Fig. 7’s note, and the IDs in Fig. 5 were added in main text. Please refer to the section 4.3, Fig. 5, and Fig. 7 in the revised manuscript.

Issue #14:  Lines 336-347 this part of the discussion seems to me very general. I would suggest the authors to improve their discussion by focusing more on the results of this work.Maybe it could be better to compare this results with similar studies. Discuss more deeply the evaluation of the impact on species, since it is an investigation that requires more complex analysis and describe if there are any limitations in this sense for this study.

Response: Revised.  We revised Section 5.3 to compare this study with previous studies and discuss our shortcomings and future directions on the evaluation of the impacts on species. In revised Section 5.2, we identified the major sources of LP for nature reserves to promote effective measures for controlling effects of LP on biodiversity conservation. Please refer to Section 5.2 and Section 5.3 in the revised manuscript.

Round 2

Reviewer 1 Report

Review report

Manuscript: “Spatiotemporal patterns of light pollution on the Tibetan Plateau over three decades”

Journal: Remote Sensing.

The reviewer thanks the authors for their comprehensive revision. Most of comments and questions have been adequately addressed and clarified. The changes improve the quality of the manuscript. However, there are still some points that need clarification.

1. As authors have changed the land use dataset (instead of using GlobaleLand30) for their reanalysis in the corresponding section. It is recommended to show ESACCI global land cover data in Figure 1 instead of GlobaleLand30.

2. Figure 2 (flowchart) should be revised, because we cannot see how Land cover data contributes to this research generally.

3. In sections 3.2 and 4.1, as my understanding, the authors extracted LP area and compared it with different factors. While your LP analysis lasts during 1992-2018 at seven milestones (1992, 1995, 2000, 2005, 2010, 2015, 2018). So I am wondering what each data point in Figure 3 reflects (year???) since the number of points here is larger than the number of milestones.

Besides, during the first round, I recommended adding a trendline in Figure 3. I understand that the authors only use a correlation coefficient to validate LP data. However, a 1-1 line somehow is good to tell how well data fits.

4. It is interesting that we currently do not have a specific threshold for LP to evaluate its influence on nature. The authors even can state it as a research need in the future. Besides, even authors have cited some references related to the “3% criteria”. However, it seems to be unreasonable. The reviewer recommends using an intermediate value – it may be general mean/median values for every year of nighttime light where LP occurs, and using it as a proxy to evaluate.

5. Figure 6 is okay. Yet, a table here may be more suitable to show the proportion of wetland and ice – it also implies important messages.

6. The reviewer appreciates that the authors have given supplementary files along. However, I meant the boundary of the general map of ecoregions, national parks should show in supplementary as complete maps. It is fine. The authors do not need to attach them as separate files.

Besides, please double-check the order and numbering of all Figures and Tables, as well as in the corresponding text, as you revised them, e.g., Figure 3 on page 8 and Figure 2 on page 9.

Author Response

Point-by-Point Response

We would like to express our respect and gratitude to the editor and anonymous reviewers for their valuable comments on improving the quality of this manuscript.  We have carefully considered all the points raised by them. The following is our point-to-point responses in the order of their comments. Both a clear version and a “track change” version of the manuscript has been submitted. Our point-to-point responses are as follows.

Reviewer #1

Issue #1: As authors have changed the land use dataset (instead of using GlobaleLand30) for their reanalysis in the corresponding section. It is recommended to show ESACCI global land cover data in Figure 1 instead of GlobaleLand30.

Response: Revised. Please refer to the revised Fig. 1.

Issue #2: Figure 2 (flowchart) should be revised, because we cannot see how Land cover data contributes to this research generally.

Response: Revised. Please refer to Fig. 2 in the revised manuscript.

Issue #3: In sections 3.2 and 4.1, as my understanding, the authors extracted LP area and compared it with different factors. While your LP analysis lasts during 1992-2018 at seven milestones (1992, 1995, 2000, 2005, 2010, 2015, 2018). So I am wondering what each data point in Figure 3 reflects (year???) since the number of points here is larger than the number of milestones.

Besides, during the first round, I recommended adding a trendline in Figure 3. I understand that the authors only use a correlation coefficient to validate LP data. However, a 1-1 line somehow is good to tell how well data fits.

Response: Clarified and revised. From the GHNTL dataset, we obtained annual nighttime light data from 1992 to 2018. To highlight key messages, we selected seven milestones of data for analysis. However, in order to fully evaluate the accuracy of the data, we use all data for accuracy evaluation. We added the explanation in Section 3.2 and Section 3.3 to avoid confusion in the revised manuscript. In addition, the 1-1 lines were added in Fig. 3

Issue #4:  It is interesting that we currently do not have a specific threshold for LP to evaluate its influence on nature. The authors even can state it as a research need in the future. Besides, even authors have cited some references related to the “3% criteria”. However, it seems to be unreasonable. The reviewer recommends using an intermediate value – it may be general mean/median values for every year of nighttime light where LP occurs, and using it as a proxy to evaluate.

Response: Revised and clarified. This point was stated in the last sentence in Section 5.3 in the revised manuscript. The “3% criteria” was only used in Section 4.3 and Fig. 5. Since this part mainly focused on the LP in 2018, we used the intermediate value (3%), i.e., the proportion of LP to the entire TP in 2018, as a proxy to find the regions with LP level higher than the average level of LP on the TP in such year. To avoid confusion, we revised Section 4.3 and Fig. 5. Please refer to the revised manuscript.

Issue #5: Figure 6 is okay. Yet, a table here may be more suitable to show the proportion of wetland and ice – it also implies important messages.

Response: Revised. The table was added. Please refer to Table 2 in the revised manuscript.

Issue #6: The reviewer appreciates that the authors have given supplementary files along. However, I meant the boundary of the general map of ecoregions, national parks should show in supplementary as complete maps. It is fine. The authors do not need to attach them as separate files.

Response: Clarified and revised. We deleted the supplementary file, and the boundaries of all ecoregions and all national parks can be found in Fig. 5.

Issue #7: Besides, please double-check the order and numbering of all Figures and Tables, as well as in the corresponding text, as you revised them, e.g., Figure 3 on page 8 and Figure 2 on page 9.

Response: Checked. The errors were resulted from track-change model.

Reviewer 2 Report

In my opinion, the authors greatly improved the manuscript. My congratulations to the authors for their excellent work and good luck for the future.

Author Response

Thanks for the reviewer's advice!